# Bio-Based Plastics Production, Impact and End of Life: A Literature Review and Content Analysis

Halayit Abrha [1,2], Jonnathan Cabrera [1], Yexin Dai [1], Muhammad Irfan [1], Abrham Toma [1], Shipu Jiao [1] and Xianhua Liu [1,*]

1    School of Environmental Science and Engineering, Tianjin University, Tianjin 300354, China; halitgebre@tju.edu.cn (H.A.); j.cabrera.10@hotmail.com (J.C.); daiyexinf@163.com (Y.D.); irfan@tju.edu.cn (M.I.); abrhamtoma16@tju.edu.cn (A.T.); jsp@tju.edu.cn (S.J.)
2    Eritrea Institute of Technology, Mai-Nefhi College of Science, Maekel 12676, Eritrea
*    Correspondence: lxh@tju.edu.cn; Tel./Fax: +86-22-85356239

**Abstract:** The accumulation of plastic wastes is one of the most widely spread problems affecting the environment. The reality that plastics can be made from renewable resources and degrade naturally has prompted academics to think outside the box to develop "better for the environment" items. In this paper, a bibliometric analysis of the scholarly publications related to bio-based plastics within the last 20 years is presented. Annual progression, geographic and research area distribution, and keyword co-occurrence were all examined. Six distinct clusters emerged from keyword analysis, which were further categorized into three directions: production to marketing; impact on the environment, economy, and society; and end-of-life (EoL) options. The major focus was on how to counter the weaknesses and challenges of bio-based plastics and take opportunities using the inherent advantages of bio-based plastics. Comprehensive studies regarding the impact of bio-based plastics on the environment, economy and social sustainability are still deficient. Although there are many promising innovations in this area, most of them are at the research stage. The benefits of bio-based plastics and better EoL options can be enjoyed only after increased production.

**Keywords:** bio-based plastics; bioplastic; biodegradable plastics; renewable; bibliometric analysis; end-of-life (EoL)

## 1. Introduction

Ease of mass production, convenience of use, and low cost, among other characteristics, have made plastic production and use extraordinary, surpassing most other manufactured materials. It has been estimated that since the beginning of mass production of plastics in the 1950s, plastic production rate has accelerated so rapidly that total production has reached 8.3 billion metric tons—most of it in disposable products that end up as trash, either in landfills or the natural environment [1]. According to studies, plastic waste entering the world's aquatic ecosystems could reach 90 million Mt/year by 2030 [2], up from 8 million Mt in 2010 [3], and an estimated 12,000 million Mt of plastic waste will accumulate in landfills or the environment by 2050 [1] if waste generation continues with no improvement in its management. Additionally, as the vast majority of monomers used to make plastics are derived from fossil hydrocarbons, most commonly used plastics are not degradable and accumulate rather than decompose in the areas they are disposed in. They persist long after their intended use and pose a continuing threat to the environment—both aquatic and terrestrial [4,5].

The stability of polymers against environmental factors, chemicals, and microorganisms has challenged society with the accumulation of plastic waste and its management worldwide. Plastic use and littering are at an all-time high, exposing an increasing number of living species, including humans, to plastic ingestion and its hazardous effects [6,7]. In light of this, the growing awareness of plastics' ecological impact has inspired the search

for more sustainable products and production technologies, end-of-life (EoL) options, and policy amendments as solutions to the increasing amount of plastic waste in the environment. Application of renewable resources, biodegradability, and energy saving are among the top criteria expected of a sustainable product such as bioplastic.

Bioplastic is an umbrella term used for plastics that are bio-based, biodegradable, or both [8]. The production of bioplastics, despite their very small percentage among overall plastic production, is growing at a rate of 10% per year, representing around 10–15% of the entire plastics market [9]. Furthermore, their application is expanding and they are becoming increasingly popular in research and the economy as construction polymers, i.e., polymers that will later constitute the structural part of a finished product; functional polymers such as inks, adhesives, and coatings; or simply as a performance enhancer. The bio-based portion of these plastics is fully or partially made of biomass and may or may not be biodegradable [8]. Made of renewable resources, bio-based plastics are expected to be viable alternatives to fossil sources while still yielding a product that provides similar benefits to traditional plastics and has environmental advantages.

Yet, bio-based plastics, as with any new product entering a competitive market, face challenges, such as prohibitive production costs, low production efficiency and poor mechanical properties, consumer apathy, small-scale production, and incompatibility with existing production and recycling processes [10,11]. There is also heated argument over their land-use competition with food production and their benefit to the environment compared to their counterpart, conventional plastics [12–14]. Several researchers are making efforts towards improving bio-based plastics' features as consumer goods and environmental friendliness, alternative production pathways, degradation rate in the environment and dedicated facilities, impact on the environment, and their valorization.

Various authors have thoroughly examined the most commonly used bio-based polymers, their applications, and breakthroughs in alternative feedstock. Among the highly cited recent reviews worth mentioning are a review of two novel approaches, catalytic and biocatalytic depolymerization of lignin [15], advances and characteristics of bio-based polymers in different applications, with emphasis on packaging, biomedical and food industry [16–18] and the environmental, social, and economic impact of bio-based plastics and their involvement in a circular economy [19–21]. Most improved and promising bio-based plastics are still in the research stage, and a thorough examination of research and publication trends in bio-based plastics research will greatly benefit new researchers in the field as well as potential investors. As the arena of bio-based plastics research has expanded significantly, a contemporary comprehension of the literature, its contributors, and the major research questions and findings is critical as well.

Bibliometrics analysis has been used to analyze the impact of research outputs using quantitative measures and complements the qualitative indicators of research impact [22]. The current review paper examines the research direction and recent advances in bio-based plastics, emphasizing biodegradable bio-based plastics, using a bibliometric analysis of scientific publications published since 2002. The main objectives of this work are to analyze the dynamics of the research area and the relationship between the major contributors, evaluate studies on production, performance, and EoL management options, and identify hot topics, major drawbacks, and plausible solutions in the field. The paper's findings are organized as follows. First, the most influential authors, journals, and countries in this field were identified using descriptive analysis, after which the prevailing thread was looked into using bibliometric and network analysis, by first dividing the literature into clusters based on citation and keyword analysis, and then performing content analysis on each cluster. Finally, hot spots in the area are identified, gaps in the literature are discussed, and future research directions are suggested.

## 2. Materials and Methods

### 2.1. Search Strategy and Data Collection

An advanced search was performed in the Web of Science (WoS) core collection on 16 February 2022, using the string TS = ("bio-based plastic*" OR "bio-based polymer*", "biobased plastic*" OR "biobased polymer*"), with further refinement based on language (English) and document type (articles and review papers). The titles and abstracts of the final remaining documents were screened manually for relevance. Data on author names, publication year, subject category, journals, and keywords were also collected for further analysis, and the bibliometric information was exported from the Web of Science database for analysis in VOSviewer (version 1.6.16). developed by Centre for Science and Technology Studies of the University of Leiden, the Netherlands (vosviewer.com).

### 2.2. Data Analysis

The total number of items identified as scholarly output and the total number of times the publications have been cited were calculated for each year (2002–2021) and most contributing countries and journals in the different research categories. Bibliometric network maps based on co-authorship and citation of retrieved publications were constructed and visualized using VOSviewer. The number of co-authored documents determined research collaboration between different countries. The nodes represent the countries and journals, and links' strength between two nodes indicates the number of publications the countries and journals represented by the nodes have co-authored in common. Citation analysis, an evaluation of the research performance of scholars [23], was conducted by making further distinctions between direct citation relations, co-citation relations, and bibliographic coupling relations. Co-occurrence of author keywords was another bibliometric method used to map the research field. The total strength of co-occurrence links with other keywords was calculated and the network between the keywords with the greatest total link was mapped. Finally, the identified clusters were further categorized into three areas and the different hot issues addressed by authors were thoroughly discussed.

## 3. Results and Discussion

A total of 1874 research papers related to bio-based plastics in the years 2002 to 2022 were retrieved from Web of Science. Articles and review papers in the English language, representing 90.66% (*n* = 1600) of all the document types, were selected for analysis. Further analysis of the retrieved reports revealed that 5618 authors, 393 journals and 85 countries had contributed publications of research papers in 81 subject categories.

### 3.1. Descriptive Analysis

3.1.1. Publication Analysis by Year, Country and Journal

Descriptive analysis showed that there was a growth in publication, with most of the publications being in the last quarter of the analysis period. Figure 1a depicts the yearly progression of publications in the past 20 years. The years 2018–2021 alone produced more than half (58.7%) of all research articles published, with 20.83% in 2021, demonstrating a growing concern in plastic pollution and interest in a sustainable product with a wide range of uses. The economic and environmental impact of the outputs outside of academia and a growing fund could have also contributed to the increasing publication. The Chinese National Funding Agency is the principal funding agency, followed by the European Union, which altogether fund almost a fifth of all publications (17.8%).

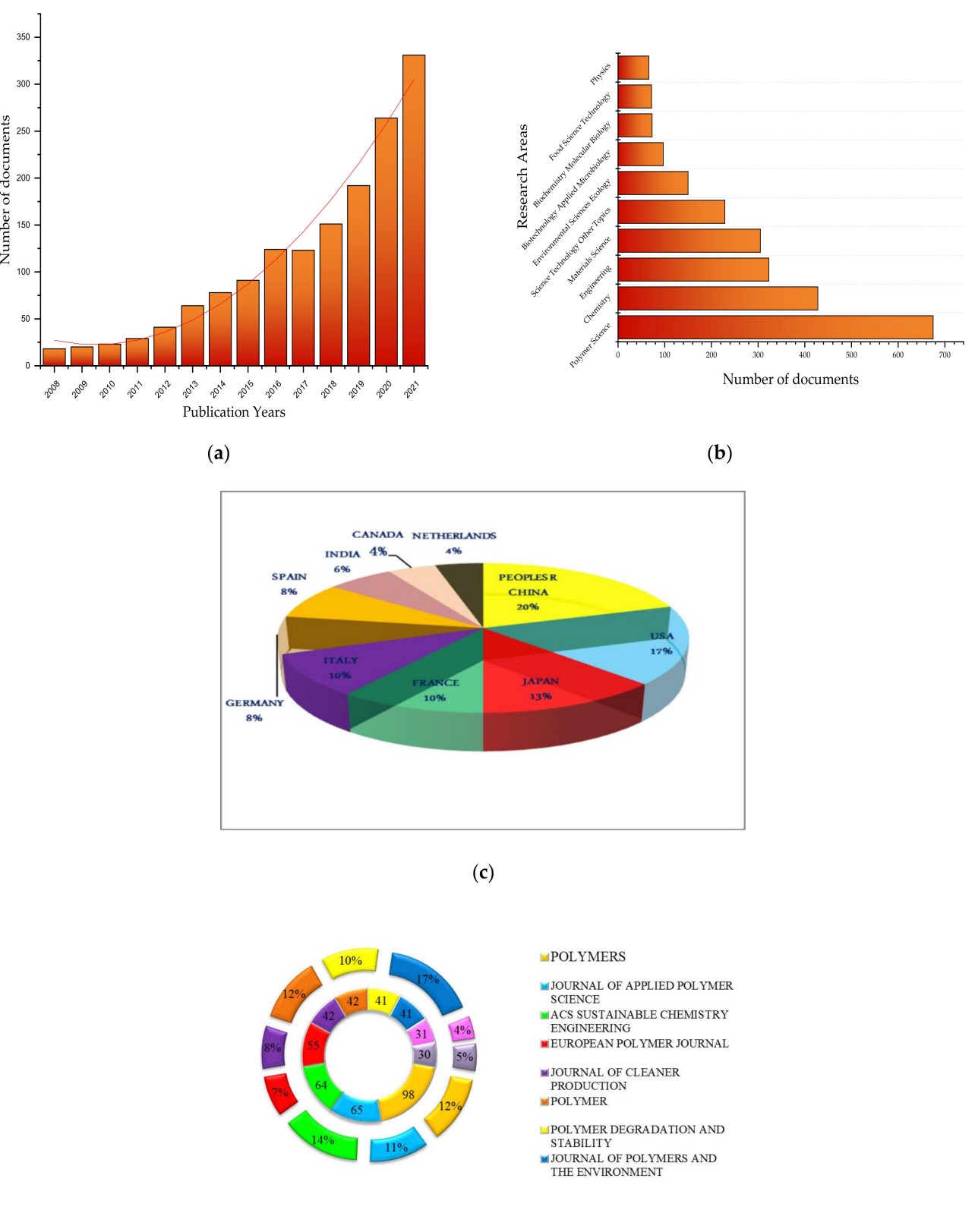

**Figure 1.** (**a**) Annual number of publications on bio-based plastics. (**b**) Number of publications in the main research areas. (**c**) Share of total publication by top ten countries. (**d**) Number of publications by top ten journals and citations obtained (citation of top ten journals add up to 26.21% of total citation).

The retrieved papers on bio-based plastics are currently undertaken by a total of 85 countries/regions, with most of the publications produced by China (17.31%), followed by the US (14.69%) and Japan (10.94%). "Polymers" is the leading journal with a share of 6.5% of total publication and the highest global distribution (40 countries/regions), and "ACS sustainable chemistry engineering" follows with a share of 4.3%. The contributing journals in the field of interest have received a total of 43,626 citations, with the top ten contributing journals accounting for more than a quarter of all citations. The most contributing countries and journals and their citation shares are depicted in Figure 1c,d.

### 3.1.2. Research Area Analysis

The journals' global and research area distribution was another analysis performed to learn the dissemination of knowledge contributed by the publications. Authors from just five countries, China, the US, Japan, France, and Italy, published more than half of the evaluated papers, with European countries and China showing dominance in the leading journals most likely due to their efforts to meet energy conservation and emissions reduction strategies and bio-economy policies. Table S1 in Supplementary Materials shows the number of countries participating in leading journals' publications. Additionally, the topic of the search was studied in 46 research areas, 42.18% of which was in the research areas of polymer science (which also embraces papers from material science), engineering, chemistry, biochemistry, molecular biology, and other fields. In this particular research area, most of the papers focus on designing, fabricating, and characterizing innovative products as they are related to a safer and sustainable environment. Preceded by environmental science and biological fields, other important areas such as agriculture, energy, metallurgy, and food science also showed a growing interest in bio-based plastics. The impact of the dominant journals, total citations, global distribution, and the number of publications in each research area is shown in Figure 1b.

### 3.2. Bibliometric Analysis

### 3.2.1. Co-Authorship: Countries and Authors

Collaboration between authors and countries was examined by country of origin. Figure 2a depicts the cooperation of different countries by network analysis. France has co-authored the most (with 39 countries) with a total link strength of 106, followed by the US, which co-authored with 35 countries. China, though with fewer co-authored papers than both France and the US, has the highest number of published papers, 108 of which are co-authored with 29 countries. Moreover, China's collaboration with other countries showed an exponential increase annually from 1 in 2012 to 16 in 2021. On the other hand, Bikiaris, Dimitrios N. has coauthored the largest number of documents ($n = 27$), with 40 authors, followed by Papageorgiou, George Z ($n = 17$). Table S2 in Supplementary Materials shows the top co-authoring countries and authors.

### 3.2.2. Citation Analysis: Documents, Journals and Authors

Citation relation patterns of publications, journals and countries were extracted and analyzed. The document with the highest citation was published by Mohanty Amar K. as first author, in 2002 in Journal of Polymers and the Environment, while the author with the highest total citation ($n = 1678$) in the analyzed topic was Caillol Sylvain. The next highly cited ($n = 988$) paper, by Laurichesse, Stephanie, was published in 2014 and has the highest average per year citation ($n = 109.78$). Among the 187 documents with total citation of $\geq 50$, 138 documents cite each other, with a total of 249 links.

On the other hand, with only ten review articles (6 of which are highly cited papers) in 10 years, "Progress in Polymer Science" was cited more than any of the 393 journals involved (n = 3182), followed by "Journal of Polymers and the Environment" ($n = 1891$), while "Polymers" showed the highest citation relationship. Moreover, there existed a variation in citations between the publication years. The research papers published in 2013 and 2016 are the most cited papers with similar citation numbers ($n = 5054$ and $n = 5049$,

respectively). However, annual citation analysis favors older articles, as most citations per year peak only four or five years after publication, and it is not good metrics to compare among publications. The citation relationship among the highly cited journals is visualized in Figure 2b.

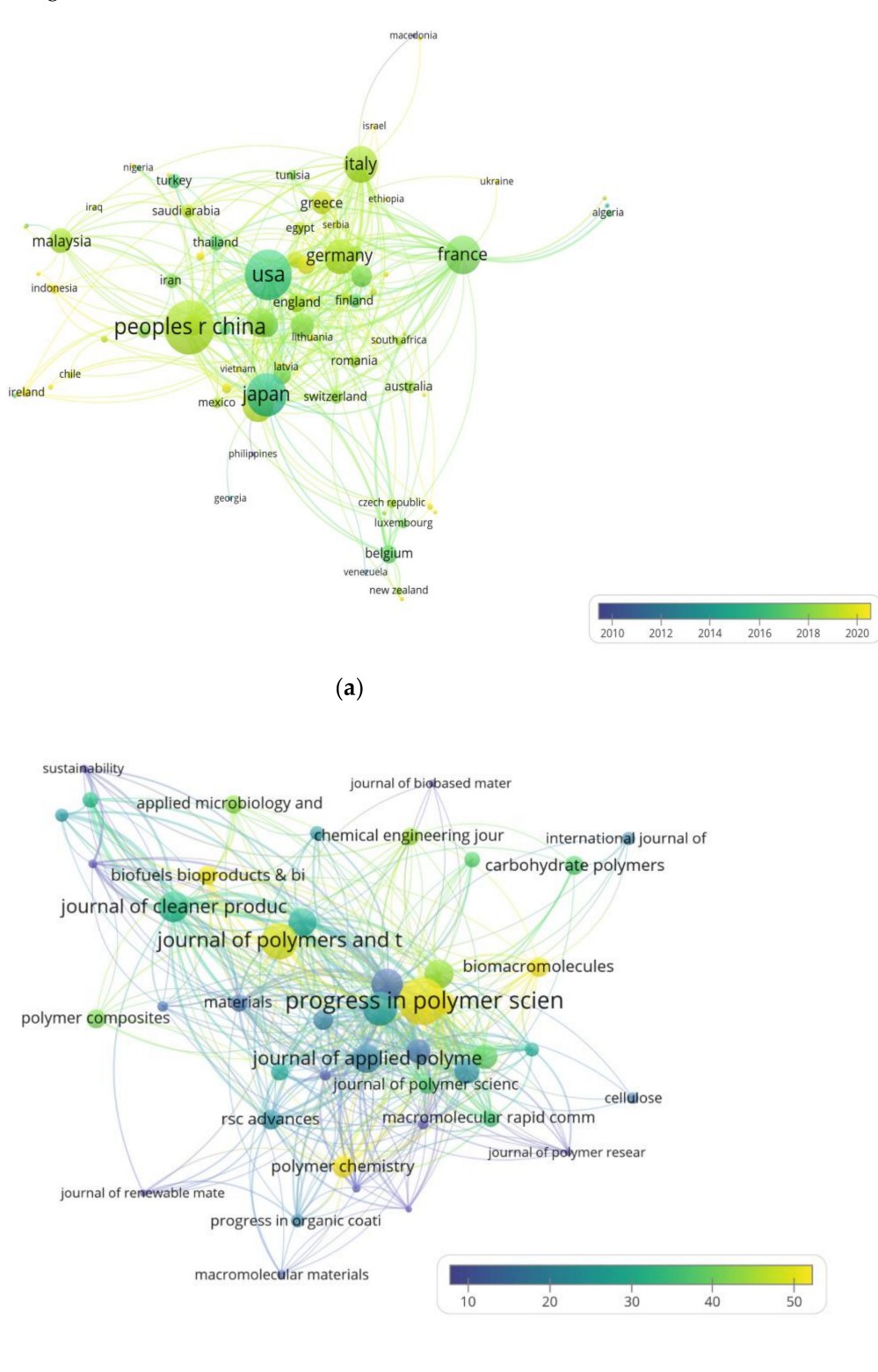

(**a**)

(**b**)

**Figure 2.** Network visualization of (**a**) countries co-authorship and (**b**) citation relationship of journals.

The country citation study reveals that the US has the most citations (*n* = 10,847) despite China having the highest number of publications. However, Chinese papers are relatively newer in achieving as high a degree of citation as those of the US. More information on co-authorship and citation relatedness can be seen in Table S3 and Figure S1 of Supplementary Materials.

### 3.2.3. Keyword Analysis

Keyword and cluster analysis were used to identify the major study direction, during which keywords with similar meanings (for example, plural and singular forms of the same word) were combined and generic terms like "bio-based plastics" were excluded. With ten times the minimum number of keyword occurrences, 54 out of 3723 author keywords were classified into five clusters. As shown in Figure 3, the terms "poly(lactic acid)" and "mechanical properties" are the first and second biggest nodes, indicating the most studied bio-based plastic and the chief objective of the research area.

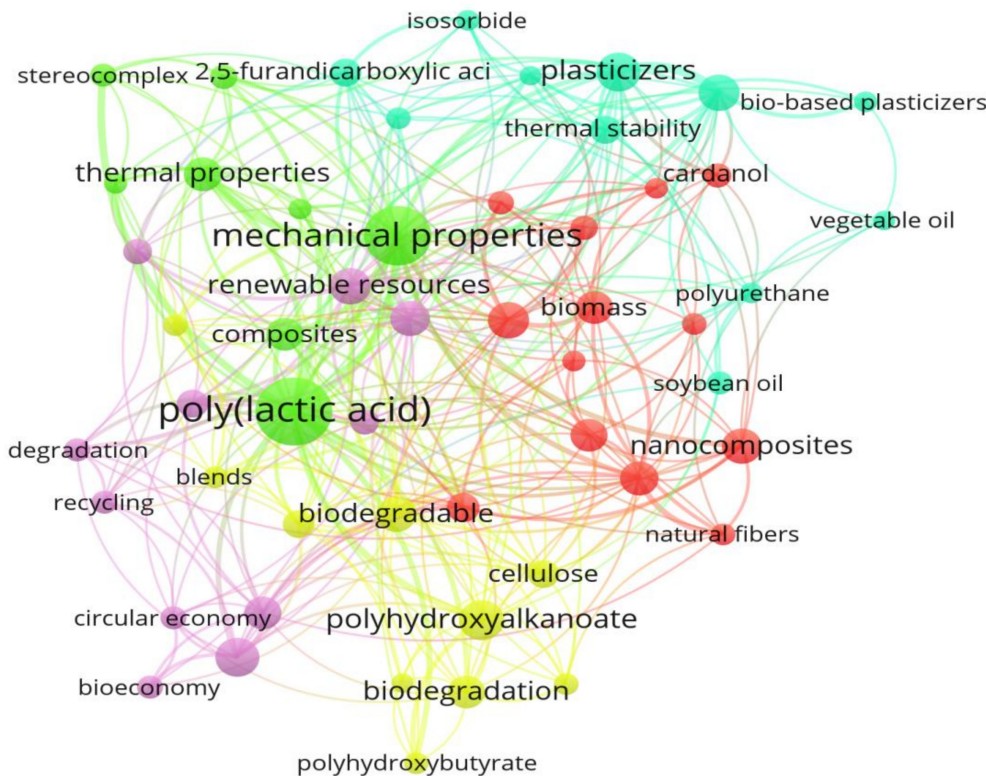

**Figure 3.** Network visualization of co-occurrence of keywords in bio-based plastic research.

### 3.3. Content Analysis

The five clusters in the network visualization were further categorized into three areas based on their objectives. The first category includes studies focusing on the production and marketing of bio-based plastics. They include developing and optimizing production processes and substrates with cost-effective, environmentally friendly, and energy-efficient techniques for generating bio-based plastics with improved applicability and features (degradability, resistance, crystallization rate, optical purity, etc.) and studies about market trends. Modification of bio-based plastics for improved sustainability and functionality and integration of bio-based plastics in the bioeconomy and circular economy is the most trending topic in the recent publications. The second category includes research works that emphasize the impacts of bio-based plastics and environmentally and economically viable EoL options. They comprise different investigations of bio-based plastics' impact and fate in the natural environment. The studies in the third category are mainly biodegradation of bio-based plastics and optimization of degradation conditions in composting and anaerobic

digestion (AD) facilities. Co-digestion, physical and chemical pretreatments, enhancement of compost/AD products (compost/digestate, biomethane) are the major areas of their focus. PHA/PHB, 2,5-furandicarboxylic acid PBS are the most frequently studied bio-based plastics, following PLA, and packaging the major application in which most researchers are interested. The research directions identified by the clusters are further categorized into three areas based on the life cycle of bio-based plastics: production and marketing; environmental, economic and social impact; and studies focusing on the fate of bio-based plastics if they are correctly disposed of. The different categories are exhaustively discussed below.

### 3.3.1. Production and Marketing

Production of bio-based plastics and incorporation of bio-based materials and their derivatives as monomers into conventional plastics have significantly expanded the content of natural products in polymer materials. The potential benefit of bio-based plastics in relying on renewable resources, has given good grounds for expecting a reduction in the global littering problem and environmental impact of products. Biodegradable bio-based plastics have the merit of being degraded under mild conditions and can be converted into their monomers and value-added products due to microbial action [24]. Currently, the leading aim of bio-based plastic production is to develop an environmentally sound, economically viable product with competitive quality as conventional plastics. The publications in this category focus on the production of bio-based plastics: production efficiency, product enhancement, sustainability and conservation of resources.

Though bio-based plastics are used in an increasing number of markets, their upscaling and commercialization are hampered by high production costs and low yield. These problems arise from substrates, microbial strains, or industrial processes and extraction methods. Studies show that feedstock and chemicals required to manufacture bio-based plastics account for about half the total cost, making their price two to three times higher than conventional plastics [25,26]. The type of the feed stock greatly affects the cost as it is associated with the type of production route and the extraction of raw materials and chemicals at an early stage of the production process [27]. The process route of polylactic (acid) (PLA) production based on corn grain is shown in Figure 4. Some research works were able to utilize feedstock that is easily available, cheaper, and environmentally friendly as an alternative to expensive raw materials. Food wastes, non-food crops, agricultural residues [28], and byproduct proteins [29,30] (wheat gluten, soybean protein, egg white albumen, rice protein, blood meal protein) (second generation feedstock), and seaweed and algae [31] (third-generation feedstock) are among the trending feedstocks being employed. Related studies aiming to reinforce the structure of the plastics made from these proteins and optimize their production processes are also underway [30,32].

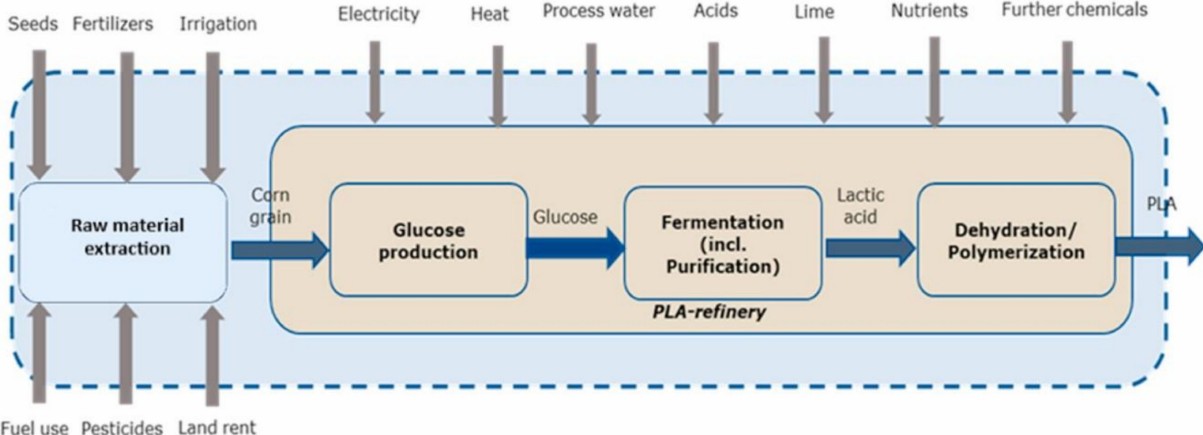

**Figure 4.** Process route of PLA-production based on corn grain [27].

Moreover, as an alternative to the expensive production method, several methods have emerged in the past few years, though only at lab scale. Pure and mixed culture and whole-cell biosynthetic systems, for example, have emerged as viable and environmentally benign, potentially shortening production processes and reducing dependence on chemicals [33]. Nevertheless, most of the promising microbial production of bio-based plastics have other issues, such as inefficient conversion to plastics, low quality, and mechanical and physical properties and monomer composition demonstrated to be strain-dependent [34,35]. In this regard, it was recommended that considering chemical interactions in addition to biological processes offers up a wide range of possibilities for developing highly efficient production strategies for target chemicals [36]. Metabolic engineering and growth optimization have also been reported to solve several such problems in synthesis pathways, substrate utilization, bio-based plastic quality, and cell morphology [37]. Apart from the weaknesses mentioned above, the limitations of bio-based plastics are also related to complicated bioprocessing associated with sterilization (high energy demand), discontinuous batch and fed-batch cultivation modes, and labor regarding their downstream processing and complex downstream separation. In light of this, studies have demonstrated that reducing the number of steps in multi-step photosynthetic polyhydroxybutyrate (PHB) manufacturing processes could save the costs that would otherwise be incurred for the purification of intermediates [38,39]. The development of technologies capable of utilizing renewable feedstock as carbon sources has to be the next strategy for improving the processes of various interlinked reactions that hinder efficiency.

Lack of a hydrolyzable functional group in most widely used plastics' C-C backbone, low surface area, high molecular weight, and crystallinity are all useful properties to obtain durable and stable polymers [40]. Conversely, the absence or limitation of these desired properties in bio-based plastics makes them less robust and flexible than traditional petroleum-based plastics, with a limited application at a large scale. Different factors, including bacterial strains involved in production, medium composition (in fermentation), carbon source, and processing methods were reported to affect the mechanical and physical properties of bio-based plastics [41,42]. Bio-based plastics must have specific properties, depending on their intended use in packaging, food services, agriculture/horticulture, electronics, automotive, consumer goods, and household appliances. In light of this, numerous studies have worked towards the modification of their mechanical (flexibility, rigidity or resilience) and physical (thermal) properties, and chemical structure, modulation of the speed of their degradation (acceleration or slowing down), and improvement of their suitability for available production methods [43–45]. Technologies such as fabrication of bio-based nanofibers through electrospinning [46] and stereocomplexation of enantiomeric PLA polymers [47] are good examples of enhancing the properties and application of bio-based plastics. Moreover, bio-based plastics were made to fit into an exciting area of innovative production of multifunctional products, such as shape memory polymers (smart polymers) with attractive application in bio-medical fields [48,49], conductive polymers in the field of electronics (reference), and sensor devices [50], the application and sustainability of which has been broadened to a great extent by incorporation of bio-based polymers.

Different strategies, including coating, blending, the addition of cellulose and other materials, and physical and chemical modification, were reported to effectively enhance bio-based plastic performance. Processing conditions and post-treatments were also reported to favor or worsen bioplastic properties [51]. In this respect, clay and organic (organoclay, nano-silica, carbon nanotube, and nano calcium carbonate, etc.), inorganic (talc, calcium carbonate and barium sulfate, etc.), and other mineral nanofillers are gaining considerable attention because of their low cost and outstanding physical and chemical properties, as well as their degradability [52,53]. Developing and identifying particles most suitable for the intended plastic properties and proper production technique, compatibility in the polymer matrix, diversification of application, and commercialization are among the hotspots in the area of these nanostructures.

Potential competition with land used for food and feed production is another threat related to bio-based plastic production, which predominantly depends on agro-based feedstock. However, this is not a major problem at present as bioplastic production only occupies less than 0.021% of global agricultural land, and a great deal of research is being conducted to diversify the availability of biogenic feedstock for the production of bio-based plastics, as well as to further develop fermentation technologies that enable the utilization of second-generation feedstock (lignocellulosic feedstock sources such as non-food crops and agricultural waste materials) [54]. Further to these efforts, other researchers are working on lessening the dependence on land-demanding crops for plastic production by switching to biomass from areas hostile to agriculture [55]. Moreover, carbon sources other than plants (shellfish [56], food waste [57], poultry feathers [58,59], sheep wool [60], etc.) are also successfully being made into fully degradable plastics with highly competitive properties and added values. However, understanding of their processing and upscaling has not yet grown. Carbon dioxide ($CO_2$) and other gases are also reported to be used as feedstock for plastic production through bio-electro recycling [61] and genetically engineered cyanobacteria (photoautotrophic bacteria) that are capable of producing polyhydroxyalkanoates (PHAs) using their self-produced glucose [62,63]. The fact that bio-based plastics can be made from $CO_2$ and at wastewater treatment plants [64] also suggests another way bioplastic production can contribute to carbon capture and waste reduction, two techniques that can mitigate environmental problems.

A growing population, urbanization, increased awareness, expanded application, as well as increased environmental concerns, are opportunities expected to drive demand for bio-based plastics and market growth in the coming years. However, little research has been conducted on the factors influencing the bioplastics market, and, most of the time, the reasons for the apparent growth of some bio-based plastics and the inability of others to scale up are only intuitively explained. According to European Bioplastics, the share of bioplastics in global plastic production will surpass 2% for the first time, and biodegradable bioplastics will more than triple in the next five years [65]. A few further studies also used models that considered learning effects, oil prices, prices for fossil-based plastic, feedstock and production costs, and price elasticity, reporting different results. This was due to an optimistic view of some variables compared to others and their complex dynamics, difficulty quantifying the influencing factors, data scarcity, and others [66]. Furthermore, efforts to identify key factors influencing market diffusion of common biopolymers by [67] the consumer and sensitivity to the consumer [68,69] are worthwhile mentioning and pursuing.

### 3.3.2. Environmental, Economic and Social Impact of Bio-Based Plastics

The impact of bio-based plastics and their biodegradability in the natural environment and dedicated degradation facilities is becoming a primary concern of a growing number of researchers. The human, environmental, and economic impacts of bio-based plastics and their biodegradability in the natural environment is mainly discussed in this section. Because of the renewable substrates they are made of, both biodegradable and non-biodegradable bio-based plastics are expected to reduce environmental plastic pollution. However, studies have revealed that none of the bio-based plastics currently in market or development are fully sustainable [70]; and despite their green origins, they do not biodegrade or require so many years to biodegrade in the natural environment [71] and are as harmful to the environment as conventional plastics [72]. They fragment into microplastics, are ingested by aquatic and terrestrial organisms, and act as carriers of toxic chemicals and microbials unless they follow a specific disposal procedure and end up in a dedicated composting facility [73]. Studies have revealed that the chemicals added to improve the materials' functionality (plasticizers, flame retardants, antioxidants, pigments, and others) induce unspecific toxicity and are prevalent in all bio-based and or biodegradable types of products [74,75]. Although the migration mechanisms of plasticizers in bio-based plastics are not fully understood, their loss to their surrounding environment has been

reported [76]. According to a recent study, bioplastics and plant-based materials were as toxic to the environment and humans as conventional plastics [77].

On the other hand, researchers have been looking into developing biodegradable, non-volatile, and nontoxic additives to improve bioplastics' mechanical, thermal, and physical properties and biodegradability to a great extent, with minimal concurrent leaching [78–81]. As was noticeable in the retrieved papers, incorporating polysaccharide or lipid-based plasticizers into conventional plastics such as poly (vinyl chloride) (PVC) has become more attractive than the traditional phthalates plasticizers and has shown competitive performance with potential in industrial application [82,83]. Combining such studies on the chemical modification of bio-based polymers with studies on potential risks of additives will aid the design of a greener product with better performance, while minimizing potential damage both to the end users and the environment.

It has been stated elsewhere that bioplastics' biodegradability is determined by their chemical structure rather than their bio-based origin. Biodegradability tests conducted in the natural environments with PHA/PHB and their blends were mostly shown to achieve the biodegradation standards in both aqueous and soil tests, as opposed to other biodegradable plastics including PLA, poly(butylene succinate) (PBS), polyhydroxyoctanoate (PHO) and their blends [84,85]. However, most degradation tests are conducted at lab scale and discrepancies between experimental and in situ studies might be observed due to sub-optimal degradation conditions in the natural environment. Such variations were also reported in similar ecosystems with differing conditions due to seasonal and compositional factors [86–88]. This necessitates extensive research on environmental factors influencing simple and composite bioplastic degradation (pH, salinity, nutrient level, microbial community, etc.) in conjunction with various environments to better understand their degradation pathways and mechanisms for producing bio-based plastics with improved degradability in the natural environment. Blending different bio-based plastics with other biodegradable or non-biodegradable plastics has also been shown to have an antagonistic and synergetic effect on their biodegradability, depending on the type of test environment [89]. Furthermore, research on bioplastics biodegradable in specific ecosystems, such as the marine environment, has recently made significant progress [90,91]. Marine biodegradable bio-based plastics with potential application in different areas have been reported in recent studies. Soni et al. [92] developed a marine biodegradable TEMPO-oxidized cellulose nano-fiber(TCNF)/modified starch membrane with eminent water resistivity, high mechanical strength and transparency, and a highly adhesive antifouling coating developed from marine bacteria [93] and biodegradable fishing gear [94]. Such studies are a huge step forward for the future of bio-based plastics and environmental research. However, simply designing biodegradable materials may not be enough; production should also include a feasible plan for recovery and treatment based on existing (or, potentially, concurrently developed) systems [95,96]. Aside from the additives and environmental biodegradability of bio-based plastics, genetically modified organisms (GMOs), persistent bioaccumulative and toxic chemicals (PBTs), and petroleum-based co-polymers used in some of the most preferred bio-based plastics are becoming an environmental concern [70].

Nonetheless, there are still many other reasons bio-based plastics benefit the environment, in terms of reduced dependence on fossil fuel resources and less greenhouse gas (GHG) emissions, a smaller carbon footprint, and faster decomposition [25]. Recently, there has been growing concern among researchers and policy makers regarding bio-based plastics' inputs and GHG emission in each stage of their life cycle and their EoL management. Life cycle assessments are timely studies that have been used to quantify the environmental impacts associated with the production, usage, and littering of commonly used materials. Using life cycle assessment methods, biological carbon content analysis can determine the carbon footprint of bio-based plastics and tests that measure the carbon isotopes of the material, which can be used to determine the bio-based content of the plastic [97]. Though many such assessments showed that bio-based plastics positively contribute to the environment [98,99], their weaknesses are also identified and improvements are suggested,

both in their production and EoL options [100]. Such improvements ought to start from the farming techniques used to grow the plants from which feedstock come from (heavy dependence on petrochemical fertilizers and pesticides) and continue to the last step of improving their recyclability. Two strategies of plastic production, switching to renewable feedstock and using renewable energy, were compared in terms of GHG emission and energy consumption [101]. The study shows switching to renewable energy achieves greater emission reductions (50–75%) and lower costs than switching to corn-based biopolymers (10–50%). However, projected emission analysis in the same study shows that reduction due to low carbon energy usage is only temporary, and long term reductions are possible by producing renewable plastics by combining advanced feedstock (switchgrass) and renewable energy. Congruence of life cycle assessment procedures while involving as many indices for environmental impact as possible was recommended as a means of including all the benefits and downsides of bio-based plastics [102].

The sustainability of a product includes its environmental, social, and economic benefits and protection of public health and environment throughout its life cycle. However, most of the sustainability assessments of bio-based plastics focused solely on a few stages of their life cycle in the environment, with sustainability of their production techniques and socio-economic impacts largely neglected, perhaps due to the limited data on their production paths and customer behavior [19]. Bio-based plastics have the potential to transform the plastic sector from the costly linear economy to a circular economy through utilizing sustainable feedstock, reducing dependence on finite resources, reducing landfill waste, and introducing new recycling and production pathways [103,104]. However, researchers show that replacing conventional plastics with bioplastics, at present, is highly unlikely due to high production costs, which according to one study would require 54% of current corn production and 60% of Europe's annual freshwater withdrawal to replace current annual global packaging plastics production [104]. On the other hand, criteria and indicators ensuring the feasibility and economic viability of mechanical and organic recycling of bio-based plastics, concerning their involvement in supporting the circular bio-economy, were proposed in two of the few assessments of their kind [105,106]. The studies show that organic recycling is an inferior EoL option, compared to material recovery through mechanical recycling, and that it should be applied only to non-recyclable biodegradable plastics. Moreover, it was suggested that innovation and financial incentives, along with integration of recirculation potential in the bioplastics design, has the potential to increase their circularity [103,105]. On the other hand, the social impact of bio-based plastics was reviewed by assuming the comparability of agricultural upstream processes and transferability of findings of the studies on social aspects of other bio-based products [19]. Inclusion of some social indicators such as end users' health and safety, feedback mechanisms, transparency, and end-of-life responsibility in the social life cycle assessment scheme for bio-based products would allow consumers to make better purchasing decisions, creating momentum on the market diffusion of a bio-based product [107]. Thorough and multidimensional-approached assessments of opportunities and trade-offs presented by bio-based plastics should be prompt to ensure the benefits of bio-based plastics and reduction/absence of their unintended consequences [108].

### 3.3.3. End-of-Life Management Options

The primary EoL options for biodegradable plastics are biological waste treatment (composting and anaerobic digestion (AD)), recycling (and reprocessing), incineration with energy recovery, chemical recycling, and landfilling. However, most bio-based plastics end up in landfills and incinerators due to littering and incorrect sorting, lack of composting infrastructure, and rejection at composting facilities due to their similarity with conventional plastics, consumer knowledge gap, costly infrastructure, and degradation rate inconsistent with other compostable wastes [109,110]. This, coupled with limited degradation in the natural environment, leaves bio-based plastics as potentially hazardous to the environment as their conventional counterparts, and their degradation mechanism understudied.

Compostability and biodegradability assessments are conducted in accordance with international standard organizations (American Society for Testing and Materials (ASTM), International Organization for Standardization (ISO), European Standards (EN), etc.) with the objective of studying biodegradation mechanisms [111], the extent and rate of their mineralization [112] as compared to the natural environment [24], factors affecting compost quality [113], and the effect of the amount of biomaterial and additives on the biodegradation profile [114]. Additional objectives are developing qualitative and quantitative biodegradation [112] and toxicity monitoring methods [115,116], identification and isolation of degrading microbes [117], and investigation of degradation effect on the composting microbial community [118]. However, incompatibility of the standards with real-world operational conditions in composting and biogas plants has been reported [116,119,120], with little action taken to improve the situation. On the other hand, it was shown that the compostability of bio-based plastics can be improved at the designing stage by means of blending additives.

Anaerobic digestion (AD) of bio-based plastics is another potentially sustainable end of life option, serving as a source of energy (biomethane) and fertilizer (digestate). A few studies have assessed the biodegradability of the most common bio-based plastics as well as operational factors and effect of pretreatment in accordance with the standards for determining anaerobic degradation of plastic materials [121,122]. Bio-based plastics show significant biodegradation, both in AD and composting environments, but only beyond the time scales established for the processes [24,119]. On the other hand, near to the maximum theoretical methane yield was achieved by alkaline, thermal, hydrothermal, gamma irradiation, steam, and combined pretreating of the plastics [123–125]. The pretreatments, by simplifying the complex structure of the bio-based plastics, increased the reaction rate and shortened the hydrolysis time, which is the rate limiting step in AD [123]. In addition, co-digestion of different organic wastes has shown a synergetic effect of an optimal carbon to nitrogen (C/N) ratio, resulting in improved quantity and quality of biogas, in addition to diverting food waste from landfills, using existing infrastructure and expertise [126]. A few studies carried out co-digestion of bio-based plastics with other wastes as municipal, food, and agricultural wastes, which resulted in modulation of the participating microbial communities, improved degradation, and biogas production [123,127]. Investigation of co-digestion of bio-based plastics, however, is only in its infancy, and more has to be done from the perspective of compatibility with the commonly co-digested wastes, digestate quality, technical feasibility, and other relevant issues.

Apart from the biodegradation of bio-based plastics in composting facilities and anaerobic digesters, recycling has captured researchers' focus as it enables production of low-carbon biopolymers from recycled materials. In this case, drop-in bioplastics, such as bio-PE and bio-PET, by fitting into the existing recycling technologies, have an advantage over the common biodegradable bio-based plastics [128]. Mechanical, chemical (hydrolysis, alcololysis, glycolysis, pyrolysis), and biological recycling have already been established and can be optimized to fit bioplastic recycling to enhance environmental and economic sustainability [25,129]. Most of these methods have been reported, with mechanical recycling outperforming all other methods, including biodegradation [130–134]. In the existing plastic recycling facilities, however, occurrence of bio-based plastics of as small an amount as 2% was reported to potentially cause contamination due to differences in melting and glass transition temperatures [135]. Bio-based plastics, like most conventional plastics, must be recycled in separate streams based on material type, and must go through separation, an infamously challenging step in the process. In light of this, installation of additional sorting equipment, such as near infrared (NIR), has been suggested [135] and efforts are being made to replace the chemical intensive separation methods with more rapid and accurate ones, such as hyperspectral analysis at the near-infrared region (900–1700 nm), as shown in one study [136].

The different waste management options offered by the renewable nature of bio-based plastics have a potential to supplement the circular economy [137] where more energy

and cost-efficient ways, such as green chemistry and enzyme-based recycling of plastic waste, are necessary [138,139]. A ground-breaking innovation by [140] showed PLA and polycaprolactone (PCL) containing nano-dispersed enzymes (<2% *w/w*) are almost fully (98%) depolymerized in days in standard soil composts and household tap water. Such findings can be used as a foundation for the next efforts, which should be directed towards searching for plastic that is inherently recyclable and has superior mechanical properties, finding more enzymes which can depolymerize other common bio-based plastics, and recovering targeted monomers which can be reused rather than wasted. Figure 5 shows how mechanical and organic recycling fit in the closed loop economic model.

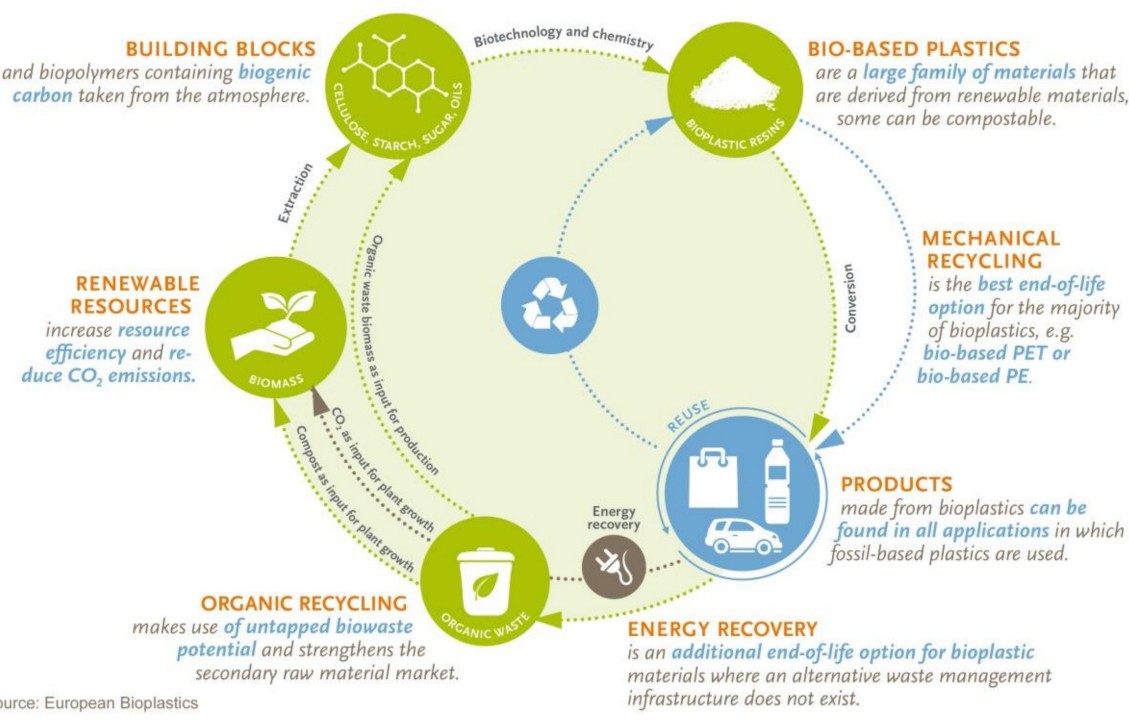

**Figure 5.** Closing the loop: life cycle assessment and life cycle economy of bioplastics. Reprinted/adapted with permission from [141]. 2016, European Bioplastics.

## 4. Conclusions

Production of bio-based plastic is a step in the right direction towards the huge environmental, economic, and social sustainability goals. It was shown that research in bio-based plastic is rapidly growing. Most of the investigations regarding bio-based plastics are focused on designing durable and sustainable products, competitive with their counterpart conventional products. Moreover, owing to their renewability and biodegradability, bio-based plastics are gaining ground in a wider range of applications than ever before. However, most of the bio-based plastics are not biodegradable in the natural environment and controversies have arisen regarding their benefit compared to conventional plastics. An increasing number of assessments have proposed integration of bio-based plastics in a bio-circular economy through use of waste-based feedstock and a zero-waste approach. Most biodegradation tests are only at lab scale, as there are few dedicated composting and AD facilities, and most bio-based plastics end up in landfills and incinerators. Bioplastics' waste stream is far below the threshold for mechanical recycling to become a beneficial EoL option.

To improve the sustainable development of bio-based plastics, the following aspects deserve further attention in near future research:

1. While designing bio-based plastics, researchers should take multiple factors into consideration, such as cost reduction and recyclability of the end product, as well as improving mechanical properties. In addition, more studies should be conducted on biodegradable additives as additives are one reason for bioplastics' recalcitrance.
2. Apart from the environmental concern, studies should focus on economic, social, and technological assessments to improve the compatibility of bio-based production with the circular economy and climate change policies.
3. To match the growing awareness and demand of bio-based plastics, the scaling up of production and promotion should be among the main objectives.

**Supplementary Materials:** The following supporting information can be downloaded at: https://www. mdpi.com/article/10.3390/su14084855/s1, Figure S1: Direct Citation relationship between countries; Table S1: Total number of publications (TNP) by leading Journals and number of Countries contributing in each journal. Table S2: Co-authorship by countries and authors with highest publication, NCC/A = Number of co-authoring countries/authors, TLS = Total link strength. Table S3: Sum of times cited (STC): documents, journals and countries.

**Author Contributions:** Conceptualization, H.A.; methodology, H.A. and J.C.; formal analysis, H.A.; investigation, H.A.; resources, H.A.; data curation, H.A.; writing—original draft preparation, H.A.; writing—review and editing, J.C., S.J., M.I. and Y.D.; visualization, H.A. and A.T.; supervision, X.L.; project administration, X.L.; funding acquisition, X.L. All authors have read and agreed to the published version of the manuscript.

**Funding:** This work was partially supported by the Natural Science Foundation of Tianjin City (Grant No. 21YFSNSN00180) and the National Key R&D Program of China (Grant No. 2019YFC1407800).

**Institutional Review Board Statement:** Not applicable.

**Informed Consent Statement:** Not applicable.

**Data Availability Statement:** Data is available from the corresponding authors upon request.

**Conflicts of Interest:** The authors declare no conflict of interest. The funders had no role in the design of the study; in the collection, analyses, or interpretation of data; in the writing of the manuscript; or in the decision to publish the results.

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
