# Peer review of "Bio-Based Plastics Production, Impact and End of Life: A Literature Review and Content Analysis"

_sustainability, doi:10.3390/su14084855_

Round 1
Reviewer 1 Report
Thank you for the interesting study.
While figure-2.a is representative, the authors are invited to review the data visualization method in Figures 2.b and 3 because the links are too dense for the reader to deduce useful information from them and it is not clear which text belong to which node. I would recommend using word-cloud figures instead of a network figures. That way the size and of the font of the text and its color would convey the information without using the links and the nodes.
Reviewer 2 Report
This paper reviews key research trends in bioplastics by using bibliometrics. The results are helpful in determining influential areas in bioplastic research. A few typos must be corrected.
- Line 338: CO2 should be written as CO2
- Line 521: "It In Light of this,..." must be corrected
- The main question raised by this review paper was "what research direction in bio-based plastics has been carried out since 2002?".
- Although the review topic is not original, this paper is still relevant to the field because it gives a visual overview of how the research direction has progressed.
- This review uses the evidence-based tool (bibliometric) to come up with visual representations, which can help readers better understand the overall research direction.
- Since bibliometrics has relied on academic papers, other types of evidence-based tools such as patents search should also be used. In addition, it is also possible to carry out interviews with diverse groups of experts or stakeholders to obtain more insights.
- The conclusion points out the findings from bibliometrics and reviewed papers.
- The following papers should be additionally reviewed and cited:
https://www.mdpi.com/2313-4321/7/2/20
https://www.nature.com/articles/s41578-021-00407-8
https://www.sciencedirect.com/science/article/pii/S2405844021020211
- The quality of all the figures must be high quality. The font sizes of all the texts must be clearly seen.
Reviewer 3 Report
Halayit Abrha describes a bibliometric analysis of the scientific publications related to bio-plastics.
The topic is of scientific interest and the investigation is appropriate. The dissertation of the major open problems related to bio-plastics is well implemented and catches the complexity of the topic. For maximizing the impact of the paper, the author may take into consideration the following aspects:
- Please increase the definition of figure 1 c and d, which are unfortunately hard to read.
- Figure 1d: “the share of total number of citation”, what is it referred to? Could the author please clarify?
- I do not fully agree with the statement at lines 155 to 159. If the consideration is based on Figure 1c, there is no reason to group this four country as “main producers” of scientific publications on biopolymers, since Italy show to have the same score as France (10%), being a much smaller country in comparison to the other named. I suggest the author to rephrase the concept reckoning the effort of Italy.
- The information available in the supporting material must be cited more carefully: please, cite every Figure of the Supporting Information in the manuscript and increase the description provided in the caption. If the figures are not relevant for improving the discussion, please erase it.
- Regarding the content analysis, I would ask the author if he would consider adding a small dissertation on the materials. Based on your key-word assessment, besides polylactic acid, which are the main bio-materials that have been studied? Which are the most recent trends?
- In some specific sections, the addition of relevant references would increase the level of the discussion. In particular, the author could take into consideration adding references in the following sections:
298 – 301: “…modification of their mechanical (flexibility, rigidity or resilience) and physical (thermal) properties, and chemical structure, modulation of the speed of their degradation (acceleration or slowing down)”.
313 - 316: “oganoclay, nano-silica, carbon nanotube, and nano calcium carbonate etc., talc, calcium carbonate and barium sulfate etc.” (In the case of organoclay, please consider the reference https://www.sciencedirect.com/science/article/pii/S0169131721004075?casa_token=mapo43wbw4QAAAAA:Kif3L8CS9OBs8HPBVgTB9Zl0XEtRBo2WbTn1s5Zzmj_0wES1SfmT2RKJtoxhsDIRNGcexQXk)
369 - 371: “plasticizers, flame retardants, antioxidants, pigments, and others” (if not for each term, at least a comprehensive one).
386 - 387
- Other less relevant comments to improve the manuscript:
English rephrasing lines 43-46, 174-176, 227, 505, 521.
Check typos at lines 167,207, 213, 338.
Check the definition of acronyms.
Reviewer 4 Report
This topic is interesting. However, the writing need be improved. This review can be considered to be published in this journal after minor revision.
Some comments:
1. The title of Figure 1a need be changed? This figure illustrates the Network visualization of countries co-authorship. Is Taiwain a country?
2. The economic impact of Bio-based plastics need be described in details, and an in-depth discussion should be made.
3. In "3.3.2. Environmental, Economic and Social Impact of Bio-Based Plastics", one related Scheme need be given.
